# The Effect of Ketoanalogues on Chronic Kidney Disease Deterioration: A Meta-Analysis

**DOI:** 10.3390/nu11050957

**Published:** 2019-04-26

**Authors:** Albert Li, Hsiang-Yen Lee, Yen-Chung Lin

**Affiliations:** 1Graduate Institute of Biomedical Electronics and Bioinformatics, National Taiwan University, Taipei 111, Taiwan; albert0325162@gmail.com; 2Department of Internal Medicine, School of Medicine, College of Medicine, Taipei Medical University, Taipei 111, Taiwan; 3Division of Nephrology, Department of Internal Medicine, Taipei Medical University Hospital, Taipei 111, Taiwan; plumlikesalt@gmail.com; 4Graduate Institute of Clinical Medicine, College of Medicine, Taipei Medical University, Taipei 111, Taiwan

**Keywords:** ketoanalogue, chronic kidney disease, estimated glomerular filtration rate, restricted protein diet

## Abstract

The effects of ketoanalogues (KA) on chronic kidney disease (CKD) deterioration have not yet been fully confirmed. To strengthen the evidence of the role of KA in CKD, PubMed and Embase were searched for studies published through February 2019. Effect sizes from ten randomized control trials (RCTs) and two non-RCTs comprising a total of 951 patients were pooled and analyzed. A restricted protein diet supplemented with ketoanalogues (RPKA) was found to significantly delay the progression of CKD (*p* = 0.008), particularly in patients with an estimated glomerular filtration rate (eGFR) > 18 mL/min/1.73 m^2^ (*p* < 0.0001). No significant change in eGFR was found when comparing a very-low-protein diet and a low-protein diet (*p* = 0.10). In addition, compared with the placebo, RPKA did not cause malnutrition (albumin: *p* = 0.56; cholesterol: *p* = 0.50). Moreover, RPKA significantly decreased phosphorous levels (*p* = 0.001), increased calcium levels (*p* = 0.04), and decreased parathyroid hormone (PTH) levels (*p* = 0.05) in patients with eGFR < 18 mL/min/1.73 m^2^. In conclusion, RPKA could slow down the progression of CKD in patients with eGFR > 18 mL/min/1.73 m^2^ without causing malnutrition and reverse CKD-MBD in patients with eGFR < 18 mL/min/1.73 m^2^.

## 1. Introduction

Chronic kidney disease (CKD) is a major public health problem worldwide. CKD without proper control may eventually lead to end-stage renal disease (ESRD). In addition, CKD might contribute to other diseases. Approximately 50 percent of adults in the United States with CKD also have coronary heart disease (CHD) [1]. Cardiovascular disease and kidney disease are closely interrelated and the disease of one organ causes dysfunction of the other, ultimately leading to the failure of both organs. This is often referred to as cardiorenal syndrome (CRS) [2].

The goal of CKD treatment is to prevent or slow further damage to the kidneys. Dietary intervention is the mainstay approach in kidney failure [3]. The principal solutes retained in CKD are the protein related products that cause hyper-azotemia, acidemia, and hyperphosphatemia; phosphate and sodium also play a relevant role in renal adaptation, causing hyperparathyroidism and extracellular volume expansion, respectively [4]. Generally, nutritional management of CKD requires balancing the intake of energy, protein, sodium, potassium, phosphorus, and fluid with biochemical markers and weight change [5].

Ketoanalogues of amino acids (KAs) are nitrogen-free analogs of essential amino acids. “Ketodiet” refers to a variety of KA and low-protein diets (LPDs; 0.6 g/kg per day) or very-low-protein diets (VLPDs; 0.3–0.4 g/kg per day), which allow a reduced intake of nitrogen while avoiding the deleterious consequences of inadequate dietary protein intake and malnourishment [6]. These diets have been proven effective in reducing renal death in selected, well-nourished, progressive CKD patients with proven diet adherence and low-comorbidity [7]. One meta-analysis [8] with stage 3–5 CKD patients who did not enter the maintenance dialysis stage proved that a restricted protein diet supplemented with KAs could delay the progression of CKD, prevent hyperphosphatemia and hyperparathyroidism, and benefit blood pressure control without causing malnutrition.

However, there is no recent research focusing on the impact of early KA intervention on nutrition status and mineral and bone disorders (MBDs). Therefore, this study sought to evaluate the effect of KA intervention on the renal function, nutrition status, and MBDs in pre-ESRD CKD patients by conducting a meta-analysis of existing trial data.

## 2. Materials and Methods

This meta-analysis was conducted according to the guidelines in the Cochrane Handbook [9]. Results are reported following the Preferred Reporting Items for Systematic Reviews and Meta-Analyses (PRISMA) statement [10].

### 2.1. Search Strategy and Eligibility Criteria

The literature search was conducted in PubMed and Embase. The search strategy used Medical Subject Heading (MeSH) terms as shown in Appendix A, and included studies published through February 2019.

Studies that met all of the following criteria were included in this meta-analysis: (1) randomized controlled trials, prospective cohort, and case-control studies; (2) CKD patients with reported estimated glomerular filtration rate (eGFR) data for various treatments; (3) intervention that compared KA with a low protein diet or very low protein against placebo.

Case reports or review articles were excluded, as well as any duplicate reports and any trials including ESRD patients.

Duplicate studies were excluded with the “find duplicate” function in the EndNote X9 reference management software (Clarivate Analytics, Philadelphia, PA, USA). Studies were initially selected based on title and abstract screening, and the full-texts were obtained for studies targeted for further review. Two authors independently identified all potentially relevant studies according to pre-specified inclusion and exclusion criteria. Any discordant evaluations were resolved by discussion and final consensus.

### 2.2. Data Extraction

Two independent researchers (A.L. and H.S.L.) independently extracted data on study characteristics, including first author’s name, publication year, location, sample size, participant characteristics, study design, and outcomes (number of participants with events in intervention group and control group) from PubMed and Embase.

The primary outcome was renal function (eGFR, creatinine, and blood urea nitrogen). Other important issues such as nutritional status (albumin and cholesterol) and CKD-MBD (phosphorous, calcium, and PTH) were analyzed as well as the secondary outcomes.

### 2.3. Quality Assessment

The methodological quality of the included non-randomized controlled trial and randomized controlled trial was assessed by the Newcastle-Ottawa Scale (http://www.ohri.ca/programs/clinical_epidemiology/oxford.asp) and Cochrane Risk of Bias Tool [11], respectively. The Newcastle-Ottawa Scale (NOS) was developed to assess the quality of non-randomized studies, and contains eight assessment items divided into three main parts: selection, comparability, and exposure (case-control studies) or outcome (cohort studies) [12]. The Cochrane Risk of Bias Tool contains: (1) random sequence generation; (2) allocation concealment; (3) blinding of participants and personnel; (4) blinding of assessment; (5) incomplete outcome data; (6) selective reporting; and (7) other sources of bias. All inconclusive results were discussed by two authors. Any further discordant evaluations were resolved by discussion with the corresponding author.

### 2.4. Statistical Analysis

The evaluation of ketoanalogues of amino acids was based on the synthesis of data extracted from the included studies. Pooling data were analyzed by Review Manager (RevMan) version 5.3, (Cochrane Community, London, UK) [13], following the Cochrane Handbook for Systematic Reviews of Interventions [9]. The fixed-effect model was used to generate forest plots for pooling data since no heterogeneity was seen between studies by using a random-effects model. The overall effects were evaluated by a z-test and calculated pooled odds ratios (OR) with 95% confidence intervals (CI). An OR > 1.0 was indicated as higher risk, and a *p*-value < 0.05 was considered to be statistically significant. A chi-square test was used to assess the heterogeneity among the included studies and *p*-values < 0.1 were considered significant. The I^2^ test was used to quantify the degree of heterogeneity and the interpretation was in accordance with the guidelines in the Cochrane Handbook for Systematic Reviews of Interventions [9], which are as follows: 0% to 40%: might not be important; 30% to 60%: may represent moderate heterogeneity; 50% to 90%: may represent substantial heterogeneity; 75% to 100%: considerable heterogeneity. Once heterogeneity was noted among the studies, random-effect model was sued. Subgroup analysis was conducted as well to determine the factors of heterogeneity. Publication biases were assessed by visual inspection of funnel plots. An asymmetric funnel plot suggested that publication bias existed. To verify the robustness of the results, sensitivity analysis was performed by excluding studies at high risk of bias one at a time and then recalculating the overall effect. All estimates and graphs were performed using RevMan 5.3 [13].

## 3. Results

### 3.1. Study Selection

We identified potential studies through searching keywords (Appendix A) in PubMed and Embase and selected eligible studies through a PRISMA flow diagram (Figure 1). Titles and abstracts were screened according to the selection process. Twenty-eight studies were excluded due to duplication; 202 studies were excluded due to irrelevance to our topic. The other 127 studies received a full-text review. Among these, 115 studies were eliminated with reasons. The remaining 12 studies, published between 1994 and 2018, were included in this meta-analysis [7,14,15,16,17,18,19,20,21,22,23,24]. Characteristics of the included studies are shown in Table 1. Of note, we divided the study led by Klahr et al. [18] into two separate studies according to the original experimental design. One with usual blood pressure, and the other one with low blood pressure. Therefore, there were 13 studies in total, instead of 12 studies, which is listed in Table 1.

### 3.2. Renal Function

Renal function indices were compared between the treatment group (low/very-low-protein diet with KA) and control group (placebo) to evaluate the effect of a restricted protein diet supplemented with KA on renal function deterioration in CKD patients

#### 3.2.1. Effects of KA on Preventing eGFR Deterioration

To compare the effect of KA on eGFR, seven RCT and one non-RCT were allocated to conduct the meta-analysis. A random model was selected due to great heterogeneity (I^2^ = 50%, *p* = 0.05) between each study.

Overall, KA significantly reversed the decrease of eGFR (mean difference (MD) = 2.74, 95% CI = (0.73, 4.75), *p* = 0.008) in CKD patients (Figure 2). A sub-group analysis was conducted to investigate whether KA plus very-low-protein (KAVLP) is more effective than KA plus low-protein diet (KALP) in preventing renal function deterioration (Figure 3). Although the mean difference of eGFR was greater in the KALP group (MD = 5.41, 95% CI = (1.74, 9.08)), a significant difference between the two subgroups was not found (*p* = 0.10). Further, since KA is efficient in protecting eGFR in CKD patients, we next assessed whether early intervention brings benefits to CKD patients (Figure 4). Four studies (394 patients) were included in the subgroup of eGFR < 18 mL/min per 1.73 m^2^, and four studies (267 patients) were included in the subgroup of eGFR > 18 mL/min/1.73 m^2^. It was found that supplemented KA benefits CKD patients with eGFR > 18 mL/min/1.73 m^2^ (MD = 5.81, 95% CI = (3.19, 8.44), *p* < 0.0001), but did not benefit patients with eGFR < 18 mL/min/1.73 m^2^ (MD = 1.87, 95% CI = (−0.08, 3.81), *p* = 0.06). The difference between the two subgroups was significant as well (*p* = 0.02).

The results suggest that KA preserves renal function in CKD patients. Although KAVLP is not better than KALP in controlling eGFR, early intervention with KA can bring benefits to CKD patients.

#### 3.2.2. Effects of KA on Serum Creatinine and Blood Urea Nitrogen (BUN) Level

The effect of KA on serum creatinine was analyzed in six studies (408 patients). Overall, a significant difference was observed between the KA and control groups (MD = 0.21, 95% CI = (0.01, 0.41), *p* = 0.04) with low heterogeneity (I^2^ = 0%, *p* = 0.71) (Figure 5). Patients within the sub-group of eGFR > 18 mL/min/1.73 m^2^ benefited from KA intervention (MD = 0.22, 95% CI = (0.01, 0.43), *p* = 0.04). However, in the subgroup of eGFR < 18 mL/min/1.73 m^2^, no significant difference was found between the KA and control group (MD = 0.18, 95% CI = (−0.44, 0.80), *p* = 0.57) (Figure 5). The overall effect of KA on serum BUN was significant (MD = 26.89, 95% CI = (10.66, 43.11), *p* = 0.001). A random model was utilized due to great heterogeneity among the studies (I^2^ = 95%, *p* < 0.00001). Subgroup analysis showed a significant decrease in BUN level in the subgroup of eGFR < 18 mL/min/1.73 m^2^ (MD = 30.40, 95% CI = (8.98, 51.83), *p* = 0.005) and the subgroup of eGFR > 18 mL/min/1.73 m^2^ (MD = 12.52, 95% CI = (5.56, 19.49), *p* = 0.0004) (Figure 6).

### 3.3. Status of Protein-Energy Wasting (PEW)

#### 3.3.1. Clinical Importance

There have been concerns of malnutrition in CKD patients due to low-protein diets. In particular, PEW is one of the strongest predictors of mortality in CKD patients [25]. The effects of KA on PEW of CKD patients were assessed by the serum albumin (g/dL) and cholesterol (mg/dL) levels [26].

#### 3.3.2. Effects of KA on Serum Albumin

To investigate the effects of KA supplements on albumin levels in CKD patients, seven RCTs and two non-RCTs were included. Overall, no significant difference was observed between the KA and placebo groups (MD = 0.02, 95% CI = (−0.04, 0.08), *p* = 0.56) (Figure 7). To assess whether renal function affects treatment outcome, subgroup analysis was conducted according to eGFR. No significant difference was found within both subgroups (eGFR < 18 mL/min/1.73 m^2^, MD = 0.00, 95% CI = (−0.06, 0.07), *p* = 0.88; eGFR > 18 mL/min/1.73 m^2^, MD = 0.16, 95% CI = (−0.05, 0.37), *p* = 0.12) (Figure 7). Next, to investigate the effects of a very-low-protein diet on the serum albumin level, we assessed the difference between patients receiving KAVLP and those receiving KALP. No significant difference was found between subgroups (*p* = 0.66) or within either subgroup (KAVLP subgroup: MD = 0.01, 95% CI = (−0.05, 0.07), *p* = 0.68; KALP subgroup: MD = 0.06, 95% CI = (−0.14, 0.25), *p* = 0.55) (Figure 8). The results suggest that the albumin level was not significantly affected by KA and that a very-low-protein diet did not significantly lower the serum albumin levels.

#### 3.3.3. Effects of KA on Serum Cholesterol

To investigate the effects of KA supplements on cholesterol levels in CKD patients, three RCTs and two non-RCTs were included in the meta-analysis. Overall, cholesterol level did not change significantly (MD = −24.13, 95% CI = (−93.68, 45.42), *p* = 0.50). Further, to investigate whether a very low-protein diet decreases serum cholesterol level in CKD patients, subgroup analysis was performed. No significance was observed between the KALP and KAVLP subgroups (*p* = 0.52). Further, no significance was observed within the two subgroups (KAVLP subgroup, MD = −29.88, 95% CI = (−109.54, 49.77), *p* = 0.46; KALP subgroup, MD = 0.3, 95% CI = (−44.61, 45.21), *p* = 0.99) (Figure 9). The results suggest that cholesterol level was not significantly affected by KA and that a very-low-protein diet did not significantly lower serum cholesterol level.

### 3.4. Mineral and Bone Disorder

#### 3.4.1. Effects of KA on Serum Phosphorus

Six RCTs and one non-RCT were included to analyze the effects of KA on serum phosphorous. Overall, KA showed significant effects on controlling phosphorous imbalance (MD = 0.26, 95% CI = (0.05, 0.47), *p* = 0.02). Subgroup analysis found that patients with eGFR < 18 mL/min/1.73 m^2^ had significantly lower phosphorous levels (MD = 0.38, 95% CI = (0.19, 0.58), *p* = 0.0001). However, in the subgroup of eGFR > 18 mL/min/1.73 m^2^, no apparent difference was found (MD = −0.02, 95% CI = (−0.36, 0.32), *p* = 0.90) (Figure 10). The results suggested that KA significantly lowers serum phosphorous levels in patients with poorer renal function.

#### 3.4.2. Effects of KA on Serum Calcium

To investigate the effects of KA on serum calcium, five RCTs and one non-RCT were included in the meta-analysis. Overall, the effect of KA on preventing calcium imbalance was not significant (MD = 0.07, 95% CI = (−0.02, 0.15), *p* = 0.11). However, subgroup analysis of patients with eGFR < 18 mL/min/1.73 m^2^ showed significantly higher calcium levels in the KA group (MD = 0.08, 95% CI = (0.00, 0.16), *p* = 0.04). No significance was observed in the subgroup of eGFR > 18 mL/min/1.73 m^2^ (MD = 0.21, 95% CI = (−0.62, 1.05), *p* = 0.62) (Figure 11). The results suggest that KA significantly increases serum calcium levels in patients with poorer renal function.

#### 3.4.3. Effects of KA on Serum Parathyroid Hormone (PTH)

Four RCTs and one non-RCT were included to analyze the effects of KA on serum PTH. Overall, KA showed no significant effects on PTH level (MD = 0.10, 95% CI = (−0.03, 0.24), *p* = 0.12). Subgroup analysis found that patients with eGFR < 18 mL/min/1.73 m^2^ had significantly lower PTH levels (MD = 0.97, 95% CI = (−0.01, 1.94), *p* = 0.05). However, in the subgroup of eGFR > 18 mL/min/1.73 m^2^, no apparent difference was found (MD = 0.00, 95% CI = (−0.07, 0.08), *p* = 0.92) (Figure 12). The results suggest that KA significantly lowers serum PTH levels in patients with poorer renal function.

### 3.5. Risk of Bias and Quality Assessment

The assessment of risk of bias is shown in Figure 13. The majority of the studies (more than 75%) showed low risk of reporting and attrition; however, low risk and/or unknown risk in selection, detection, performance, and other bias cannot be excluded.

### 3.6. Publication Bias

The funnel plot of the effect of KA on eGFR in CKD patients appeared to be symmetrical as shown in Figure 14. However, quantitative analysis of the funnel plot by Egger’s weighted regression method [27] was not conducted due to the small number of included studies. Therefore, the possibility of publication bias could not be totally excluded due to the limited power of the tests to identify a true asymmetry [9].

## 4. Discussion

Chronic kidney disease and its related comorbidities such as cardiovascular diseases, mineral bone diseases, and anemia have caused great health burdens worldwide [28,29,30,31,32]. Progression of CKD ultimately leads to the end-stage of renal disease (ESRD), which currently cannot be cured by any medical treatment [33]. Despite the rapid development of molecular medicine, medications to treat CKD have not yet been found. Therefore, the prevention of CKD progression has become one of the salient issues in prolonging the survival time for CKD patients. The current strategy for controlling CKD is relatively conservative, and aims to delay the time for dialysis and to relieve symptoms and signs caused by CKD-related comorbidities. Ketosteril^®^ is one of the KA formulations that may slow down the progression of CKD. However, solid evidence proving the efficacy of these nutritional supplements is lacking. We performed this meta-analysis of studies with different designs and types of intervention to add more evidence on the effects of KA on CKD patients.

This is the first meta-analysis to find that KA intervention in pre-ESRD patients with eGFR > 18 can prevent renal function deterioration, and in patients with eGFR < 18 mL/min/1.73 m^2^, it can alleviate CKD-MBD. These findings not only suggest the crucial role of KA in preventing CKD progression, but also imply new insights that early intervention can effectively slow down this progression.

Compared to the latest published systemic review and meta-analysis in 2015, including 9 studies and 410 patients [8], our results included both a greater number of clinical studies (10 RCTs and 2 non-RCT) and a larger sample size (951 patients in total). We not only showed that KA was effective in preventing renal function deterioration and CKD-MBD, but also revealed that KA had a different impact depending on the CKD stage. These results may provide more solid evidence to clinicians in determining when to start KA intervention and the expected benefits to the patients.

Nutrition deficiency is one of the major concerns in CKD patients. Restricted protein diets can protect the kidneys and delay CKD deterioration by decreasing albuminuria and alleviating renal fibrosis [34,35,36]. However, these diets may lead to malnutrition [35], contributing to nutrition-related comorbidities such as metabolic acidosis [37,38], hormone disorders [39,40,41,42], sustained inflammation [43,44,45], and protein energy wasting (PEW) [46,47]. A study from Modification of Diet in Renal Disease (MDRD) further suggested that in long-term follow up, a very low-protein diet did not delay progression to kidney failure, but appeared to increase the risk of death [48]. The major limitation of this study, however, was a lack of dietary protein measurements during follow-up [48]. Therefore, our study used subgroup analysis to provide evidence as to whether KAVLP is better than KALP in controlling CKD. It was revealed that albumin, cholesterol, and eGFR levels did not show a significant difference between KAVLP and KALP groups. That is, patients with a very-low-protein diet did not have a higher risk of malnutrition, but also did not receive extra benefits in preventing renal function deterioration, suggesting that a very-low-protein diet may not be necessary. There are, however, limitations in this argument, due to the indirect inference from the subgroup analysis. The results may be affected by other confounding factors. More rigorous studies such as RCTs are suggested to address the issues of very-low-protein diets, malnutrition, and mortality. Last but not least, the use of cholesterol and albumin to represent the nutrition status may be arbitrary and up for debate [25,49]. It could be true that using a comprehensive parameter, such as body mass, muscle mass, dietary intake, and nutritional scoring systems, to evaluate the nutritional status instead of simply albumin/cholesterol, is a better and more meaningful and more holistic approach [26]. However, there are still arguments in favor of simply using albumin as a nutritional marker in assessing kidney diseases [50]. In our opinion, detailed parameters, such as the percent of fat, and carbohydrate content, were not shown in most of the studies that we allocated. Therefore, it may be difficult to analyze such parameters using meta-analysis considering the limited number of included studies. Further, we believe that by simply monitoring albumin and cholesterol in CKD patients is a more practical way in daily clinical practice to assess CKD patients’ nutritional status particularly in medical-lacking or developing countries. The Kidney Disease Improving Global Outcomes (KDIGO) group defines CKD-MBD as a systemic disorder of mineral and bone metabolism caused by CKD and manifested by one or a combination of the following: (1) abnormalities of calcium, phosphorous, PTH, or vitamin D metabolism; (2) abnormalities in bone turnover, mineralization, volume, linear growth, or strength; or (3) vascular or other soft-tissue calcification [30,51]. Previous studies revealed that elevated serum phosphorous or calcium-phosphorous product, or high PTH were associated with poorer clinical outcomes and higher mortality risk in CKD patients [51,52,53,54]. Investigation of treatment for CKD-MBD was thus warranted. However, the benefit of KA on reversing CKD-MDB remained inconclusive [7,15,19,23]. Jiang et al. revealed that KA significantly decreased serum phosphorous and PTH levels (phosphorous: MD = −0.20 (−0.29, −0.11), *p* < 0.0001; PTH: MD = −2.43 (−4.75, −0.11), *p* = 0.04) but did not affect calcium levels (MD = 0.07 (−0.06, 0.20), *p* = 0.28) [8]. Our overall results, in contrast, found no significant difference in serum phosphorous, calcium, and PTH between RPKA and placebo groups. Of note, however, all three key parameters showed a significant difference in the subgroup of patients with eGFR < 18. Further investigation revealed that the mean phosphorous (mmol/L), calcium (mmol/L), and PTH (pg/mL) levels in the eGFR < 18 and eGFR > 18 groups were 1.75 vs. 1.32, 1.704 vs. 2.235, and 2.191 vs. 0.0755, respectively. Although the differences were not statistically analyzed, patients with more advanced CKD tend to have more severe hyperphosphatemia and hypocalcemia, and higher PTH levels, which corresponds to other published literature and our clinical experience that the incidence and severity of CKD-MBD is positively associated with the CKD stage. Therefore, we believe that early KA intervention is effective in reversing CKD-MBD, but the statistical difference is only seen when the severity of CKD-MBD is high.

There are several limitations to our meta-analysis. First, the number of studies and subjects included is relatively small, resulting in limited statistical power. Second, we included 12 studies comprising 10 RCTs and 2 non-RCTs, which may lead to increased heterogeneity and uncontrolled bias although some arguments state that the advantages of including observational studies into a meta-analysis could outweigh the disadvantages [55]. Third, we were unable to allocate all patient-level data. Fourth, the patients analyzed in this study belonged to CKD stage 4 to 5. Therefore, the conclusions may not be directly applied to patients with earlier CKD. Finally, the number of included studies have still been not great enough to conduct statistically meaningful analysis of publication bias. These points highlight the need to interpret the results of this meta-analysis with caution and the need for better designed, high-quality RCTs with larger sample sizes for a more precise evaluation.

## 5. Conclusions

Our findings added new evidence to expand our knowledge on the renal protecting effect of KA in CKD patients. To the best of our knowledge, it was the most comprehensive and updated meta-analysis investigating the role of KA in CKD. We found that KA had a significant role in preventing CKD progression and specified the sub-group of patients with eGFR > 18 that can truly receive an advantage from KA intervention. In addition, we determined that a very-low-protein diet may not bring extra benefit to patients. Finally, KA is effective in protecting renal deterioration (GFR reduction) in patients with advanced CKD stage (eGFR < 18), and in controlling CKD-MBD without adverse effects on nutrition and cholesterol level. However, more RCTs are needed to resolve the problems of publication bias and high heterogeneity amongst studies.

## Figures and Tables

**Figure 1 nutrients-11-00957-f001:**
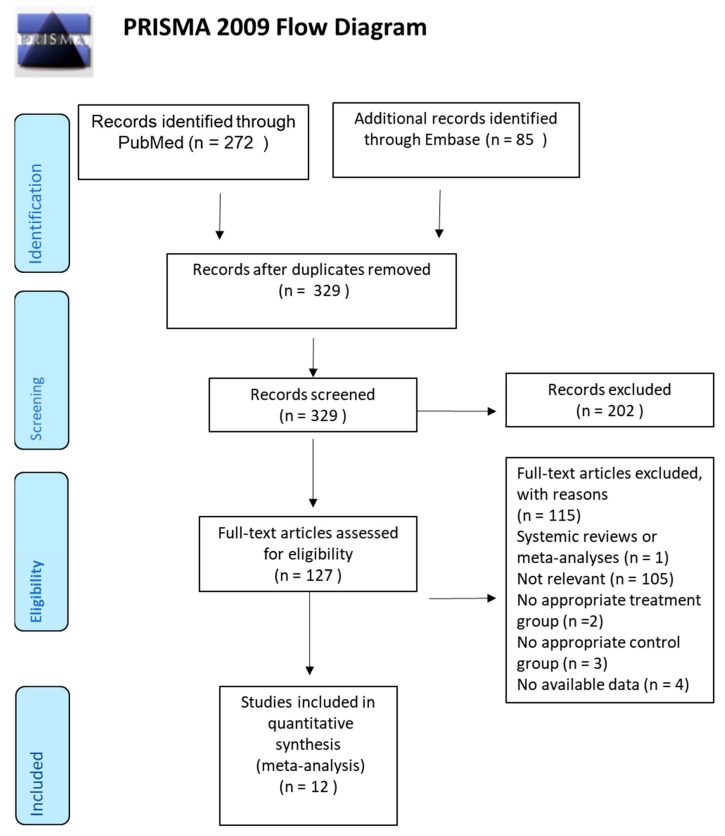
Preferred Reporting Items for Systematic Reviews and Meta-Analyses (PRISMA) flow diagram of included articles.

**Figure 2 nutrients-11-00957-f002:**
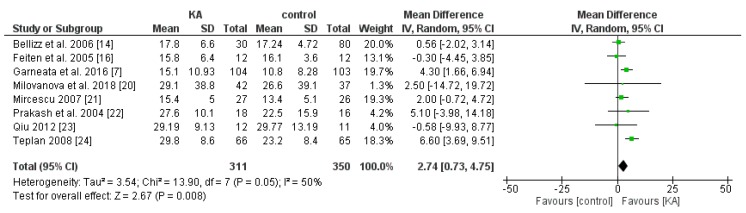
Forest plot of effects of ketoanalogues (KA) on estimated glomerular filtration rate (eGFR) in chronic kidney disease (CKD) patients, based on the random-effects model.

**Figure 3 nutrients-11-00957-f003:**
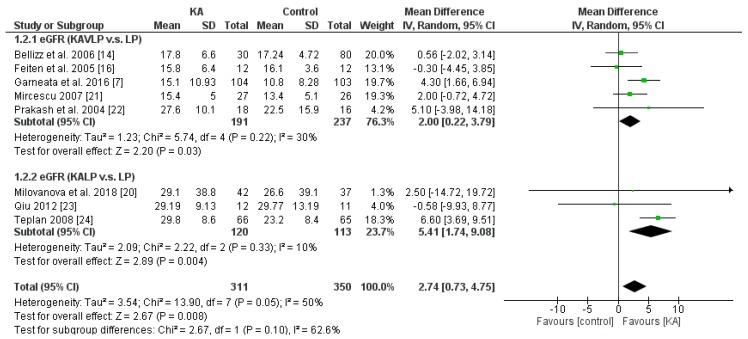
Forest plot of effects of KA on eGFR in CKD patients sub-grouped by the type of restricted protein diet. The random-effects model was utilized. KAVLP: very-low-protein diet supplemented with ketoanalogues. KALP: low-protein diet supplemented with ketoanalogues. LP: low-protein diet.

**Figure 4 nutrients-11-00957-f004:**
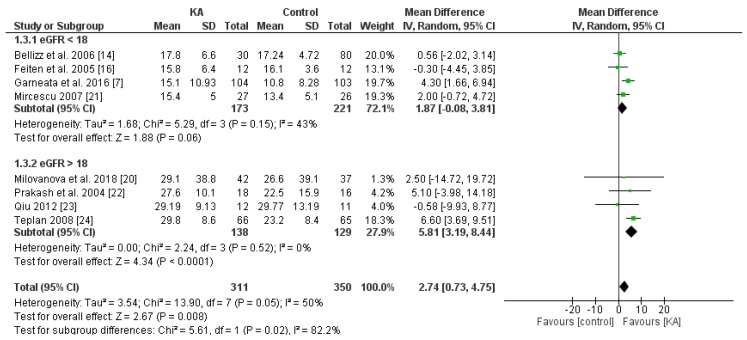
Forest plot of effects of KA on eGFR in CKD patients sub-grouped by eGFR. The random-effects model was utilized.

**Figure 5 nutrients-11-00957-f005:**
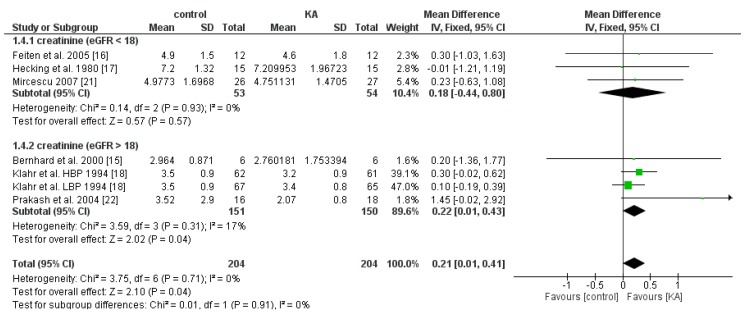
Forest plot of effects of KA on creatinine in CKD patients sub-grouped by eGFR. The fixed-effects model was utilized.

**Figure 6 nutrients-11-00957-f006:**
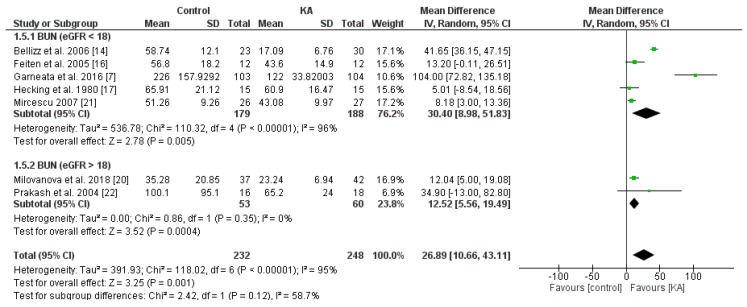
Forest plot of effects of KA on BUN in CKD patients sub-grouped by eGFR. The random-effects model was utilized.

**Figure 7 nutrients-11-00957-f007:**
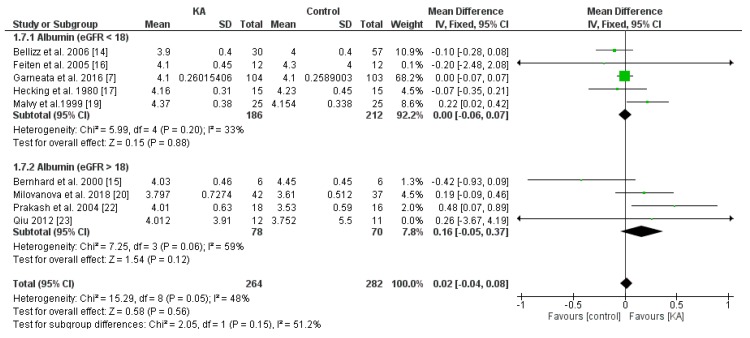
Forest plot of effects of KA on albumin levels in CKD patients sub-grouped by eGFR. The random-effects model was utilized.

**Figure 8 nutrients-11-00957-f008:**
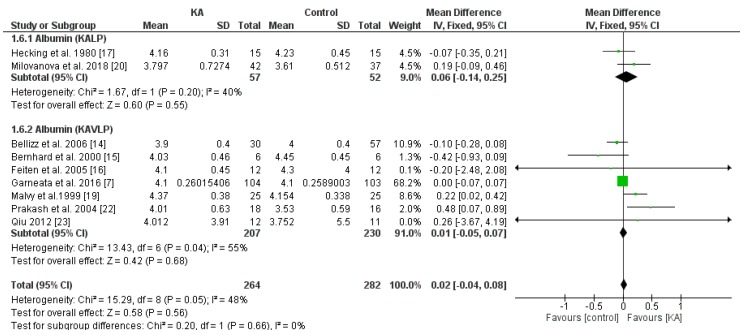
Forest plot of effects of KA on serum albumin in patients sub-grouped by the type of restricted protein diet. The random-effects model was utilized. KAVLP: very-low-protein diet supplemented with ketoanalogues. KALP: low-protein diet supplemented with ketoanalogues. LP: low-protein diet.

**Figure 9 nutrients-11-00957-f009:**
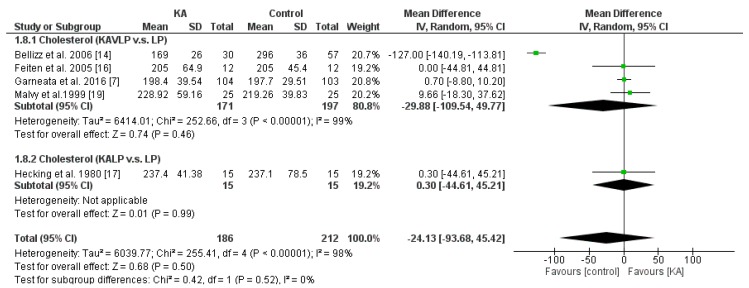
Forest plot of effects of KA on serum cholesterol level in patients sub-grouped by the type of restricted protein diet. The random-effects model was utilized. KAVLP: very-low-protein diet supplemented with ketoanalogues. KALP: low-protein diet supplemented with ketoanalogues. LP: low-protein diet.

**Figure 10 nutrients-11-00957-f010:**
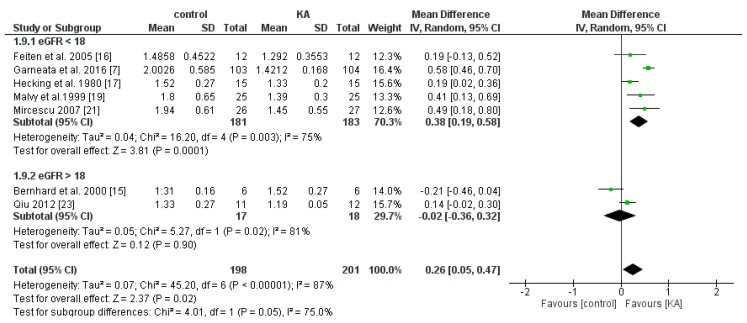
Forest plot of effects of KA on phosphorous in CKD patients sub-grouped by eGFR. The random-effects model was utilized.

**Figure 11 nutrients-11-00957-f011:**
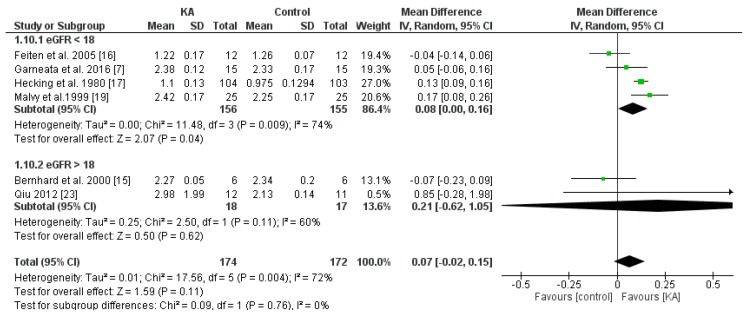
Forest plot of effects of KA on calcium in CKD patients sub-grouped by eGFR. The random-effects model was utilized.

**Figure 12 nutrients-11-00957-f012:**
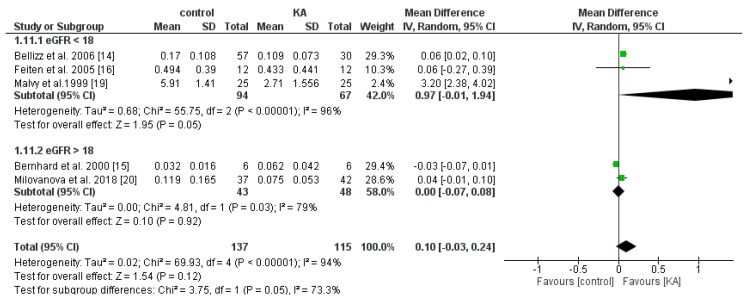
Forest plot of effects of KA on parathyroid hormone (PTH) in CKD patients sub-grouped by eGFR. The random-effects model was utilized.

**Figure 13 nutrients-11-00957-f013:**
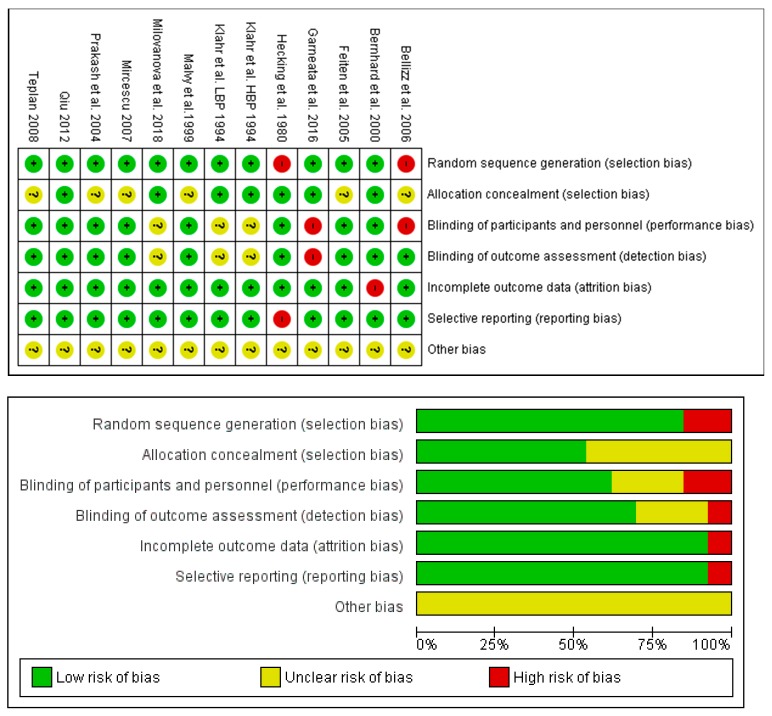
Assessment of the risk-of-bias of the studies included in this meta-analysis. (**Upper panel**): summary of the risk-of-bias. (**Lower panel**): graph of the risk of bias.

**Figure 14 nutrients-11-00957-f014:**
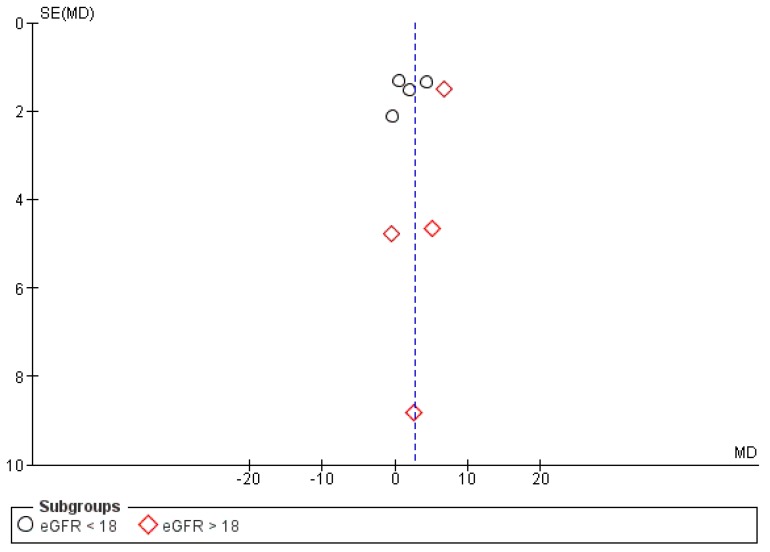
Funnel plot of the effects of KA on eGFR (mL/min/1.73 m^2^) under a random model with a 95% confidence interval.

**Table 1 nutrients-11-00957-t001:** Characteristics of the selected studies.

First Author, Year	Country	Regimen	Sample Size (Male,Female)	Treatment Duration	Study Design	Age (years)
Bellizzi, 2006 [14]	Italy	T: KAVLP (1 pill/5 kg/day) C: FD	T: 30 C: 23	3–6 months	Non-RCT	T: 58.0 ± 16.1 C: 56.3 ± 15.6
Bernhard, 2000 [15]	France	T: KALP (1 pill/5 kg/day) C: Placebo	T: 6 (4,2) C: 6 (6,0)	3 months	RCT	T: 49.5 ± 7.0 C: 39.0 ± 5.8
Feiten, 2005 [16]	Brazil	T: KAVLP (1 pill/5 kg/day) C: Placebo	T: 12 (7,5) C: 12 (8,4)	4 months	RCT	T: 49.7 ± 11.3 C: 43.9 ± 16.3
Garneata, 2016 [7]	Romania	T: KAVLP (1 pill/5 kg/day) C: Placebo	T: 104 C: 103	12 months	RCT	NA
Hecking, 1980 [17]	Germany	T: KALP (1.05 g/10 kg/day) C: Placebo	T: 15 (7,8) C: 15 (7,8)	6 weeks	Non-RCT	T: 43.7 ± 12.6 C: 43.7 ± 12.6
Klahr, 1994 (HBP) [18]	USA	T: KAVLP (0.28 g/kg/day) C: Placebo	T: 61 C: 62	18–45 months	RCT	NA
Klahr, 1994 (LBP) [18]	USA	T: KAVLP (0.28 g/kg/day) C: Placebo	T: 65 C: 67	18–45 months	RCT	NA
Malvy, 1999 [19]	France	T: KAVLP (0.17 g/kg/day) C: Placebo	T: 25 (11,14) C: 25 (10,15)	3 months	RCT	T: 53.6 ± 11.0 C: 56.0 ± 14.0
Milovanova, 2018 [20]	Russian Federation	T: KALP (0.1 g/kg/day) C: Placebo	T: 42 C: 37	14 months	RCT	NA
Mircescu, 2007 [21]	Romania	T: KAVLP (1 pill/5 kg/day) C: Placebo	T: 27 (17,10) C: 26 (15,11)	15 months	RCT	T: 55.0 ± 12.7 C: 53.6 ± 11.0
Prakash, 2004 [22]	India	T: KAVLP (1 pill/5 kg/day) C: Placebo	T: 18 (10,8) C: 16 (7,9)	9 months	RCT	T: 52.8 ± 14.1 C: 55.9 ± 17.6
Qiu, 2012 [23]	China	T: KAVLP (1 pill/5 kg/day) C: Placebo	T: 12 C: 11	52 months	RCT	T: 63.0 ± 8.9 C: 61.60 ± 9.67
Teplan, 2008 [24]	Czech	T: KAVLP (0.1 g/5 kg/day) C: Placebo	T: 66 C: 65	36 months	RCT	T: 52 ± 7 C: 52 ± 7

HBP: usual blood pressure; LBP: low blood pressure; KAVLP: ketoanalogues supplemented with very-low-protein diet; KALP: ketoanalogues supplemented with low-protein diet; RCT: randomized control trial; FD: free diet; P: placebo; T: treatment group; C: control group; NA: not applicable.

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
