# Peer review of "The Effect of Ketoanalogues on Chronic Kidney Disease Deterioration: A Meta-Analysis"

_nutrients, 2019, doi:10.3390/nu11050957_

Round 1
Reviewer 1 Report
Authors presented this important study very well. This is the first meta-analysis of 12 studies (13 studies included in Table 1, please clarify that) around the world to confirm the effects of KA on CKD patients.
Do we know the mechanisms by which RPKA diet slow down the progression of CKD in patients with eGFR > 18?
Author Response
Reviewer #1:
Q1 : Authors presented this important study very well. This is the first meta-analysis of 12 studies (13 studies included in Table 1, please clarify that) around the world to confirm the effects of KA on CKD patients.
Ans1: Thanks for your great comments. In this meta-analysis, 12 studies were included. However, it can be seen that we divided Klahr et al. into two separate studies. One with usual blood pressure, the other one with low blood pressure. We clarify this problem in page 3, line 24-26.
page 3, line 24-26…….” Of note, we divided the study led by Klahr et al. into two separate studies according to the original experimental design. One with usual blood pressure, and the other one with low blood pressure. Therefore, there were 13 studies in total, instead of 12 studies, which listed in Table 1.”
Q2: Do we know the mechanisms by which RPKA diet slow down the progression of CKD in patients with eGFR > 18?
Ans2: Thanks for your comments on this important issue. The mechanisms of KA in preventing renal function have still not been fully discovered. But effect of transamination to lower urea formation from amino acid degradation and decrease of protein-energy wasting of CKD patients by nutritional support will be possible explanations. Malnutrition is one of the strongest predictor of mortality in CKD patients (Nature Reviews Nephrology volume 7, pages 369–384 (2011)). To make this point clearer, we made a few revisions in our manuscript in page 8, line 9-11.
page 8, line 9-11…….”There have been concerns of malnutrition in CKD patients due to low protein diets. In particular, PEW is one of the strongest predictors of mortality in CKD patients. The effects of KA on PEW nutritional status of CKD patients were assessed by the serum albumin (g/dL) and cholesterol (mg/dL) levels.”
As for the reason why KA can only benefit CKD patients with eGFR > 18, we postulated that patients with advanced CKD (eGFR < 18) will have more sever comorbidities, such as CKD-MBD, cardio-renal syndrome etc., compared to moderate CKD stage (eGFR > 18). Such CKD comorbidities may act as key factors in the so-called vicious cycle, which accelerate the progression of CKD. Therefore, benefits can only be seen in patients with moderate CKD stage when such comorbidities are not severe enough to deteriorate the renal function.
Reviewer 2 Report
Albert Li et al, in “The effect of ketoanalogues on chronic kidney disease deterioration: a meta-analysis” present a thorough meta-analysis of 10 randomized and 2 non-randomized trials including a total of 943 CKD patients that passed their selection criteria to determine the potential benefits of ketoanalogues addition to low or very low protein diets on a number of health outcomes associated with the progression of CKD. The effects were subdivided in patients with advanced renal disease, eGFR<18 vs not, detecting an additional benefit in reversing CKD-related mineral and bone disease in patients with eGFR<18.< span="">
The study appropriately employs the guidelines of the Cochrane Handbook for meta-analysis and provides a detailed PRISMA checklist in the supplemental table 1. The statistical methodology is robust, and the conclusions presented are supported by the data. Moreover, the authors discuss several caveats and limitations of their study.
The following points would deserve further clarification and discussion:
· The choice of the quality assessment scale used to assess the included studies, Newcastle-Ottawa scale, needs further justification and discussion. The authors cite a critical review of the scale rather than the original publication (http://www.ohri.ca/programs/clinical_epidemiology/oxford.asp), please correct and provide further support of why this was chosen for quality assessment, over for example the Methodological Index for Non-Randomized Studies (MINORS), as this will have major effects on the results of the meta-analysis. For further critique of the scale, see Hartling et al, PMID: 23683848.
· It is unclear why a subset of studies was included in the model to assess overall effects of KA in eGFR. Why not include all studies in the overall comparison and then proceed in the subset analysis by VLP vs LP and advanced vs moderate CKD stage?
· No statistically significant effects were observed in albumin or cholesterol levels with KA supplementation. Based on this result, the authors conclude that KA supplementation has no significant effect on patients’ nutritional status. Further discussion about the percent fat and carbohydrate content of VLP and LP diets and their effect on cholesterol level (after adjusting for statin use) and cardiovascular risk, seems warranted.
· How did the short duration non-RCT study by Hecking et al (6 weeks), and the study with very low number of subjects, by Bernhard et al (N=6 per group), pass quality criteria for inclusion?
· Please provide more details about the dose and type of ketoanalogues and dietary protein amounts, in g/kg/d, used in each study. Age, gender and race distribution of the subjects studied would be very useful to include in table 1.
· For improved consistency, consider presenting KA vs control data in the same order in tables/figures throughout the manuscript. Figures 5, 6, 10, and 12 are different to the rest (controls presented before KA treated subjects in the tables), unless there is another rational behind the different format.
Author Response
Reviewer #2
Albert Li et al, in “The effect of ketoanalogues on chronic kidney disease deterioration: a meta-analysis” present a thorough meta-analysis of 10 randomized and 2 non-randomized trials including a total of 943 CKD patients that passed their selection criteria to determine the potential benefits of ketoanalogues addition to low or very low protein diets on a number of health outcomes associated with the progression of CKD. The effects were subdivided in patients with advanced renal disease, eGFR<18 vs not, detecting an additional benefit in reversing CKD-related mineral and bone disease in patients with eGFR<18.< span="">
The study appropriately employs the guidelines of the Cochrane Handbook for meta-analysis and provides a detailed PRISMA checklist in the supplemental table 1. The statistical methodology is robust, and the conclusions presented are supported by the data. Moreover, the authors discuss several caveats and limitations of their study.
The following points would deserve further clarification and discussion:
Q1: The choice of the quality assessment scale used to assess the included studies, Newcastle-Ottawa scale, needs further justification and discussion. The authors cite a critical review of the scale rather than the original publication (http://www.ohri.ca/programs/clinical_epidemiology/oxford.asp), please correct and provide further support of why this was chosen for quality assessment, over for example the Methodological Index for Non-Randomized Studies (MINORS), as this will have major effects on the results of the meta-analysis. For further critique of the scale, see Hartling et al, PMID: 23683848.
Ans1: Thanks for your invaluable comments. We cite the NOS URL in the text. It is not denying that several limitations such as low reliability between individual reviewers and poor agreement in using NOS. (J Clin Epidemiol. 2013 Sep;66(9):982-93.; AND World J Meta-Anal. Aug 26, 2017; 5(4): 80-84). However, it is also clear that NOS is one of the most popular assessment tools and provides a quick, adaptable, and validated items to appraise non-RCTs (World J Meta-Anal. Aug 26, 2017; 5(4): 80-84). Therefore, we still used NOS as the tool here to make critical appraisal of non-RCTs in our meta-analysis. Two authors (A.L and H.S..L appraised the articles independently. If inconclusive results were found, the correspondent author (Y.C.L) was consulted. The items will be discussed together one-by-one. It is believed that the disparity between each author can be very small.
Page2, line 44 to page 3, line 3………. “The methodological quality of the included non-randomized controlled trial and randomized controlled trial was assessed by the Newcastle-Ottawa Scale (http://www.ohri.ca/programs/clinical_epidemiology/oxford.asp) and Cochrane Risk of Bias Tool, respectively. The Newcastle-Ottawa Scale (NOS) was developed to assess the quality of non-randomized studies, and contains eight assessment items divided into three main parts: selection, comparability, and exposure (case-control studies) or outcome (cohort studies) [11]. The Cochrane Risk of Bias Tool contains: (1) random seuqce generation; (2) allocation concealment; (3) blinding of participants and personnel; (4) blinding of assessment; (5) incomplete outcome data; (6) selective reporting; and (7) other sources of bias. All inconclusive results were discussed by two authors. Any further discordant evaluations were resolved by discussion with the corresponding author.”
Q2: It is unclear why a subset of studies was included in the model to assess overall effects of KA in eGFR. Why not include all studies in the overall comparison and then proceed in the subset analysis by VLP vs LP and advanced vs moderate CKD stage?
Ans2: Thanks for your great comments. We completely agree your logic reasoning for the experimental design. Therefore, we presented an overall effect of KA on eGFR in Figure 2 (no subset analysis was included), which showed KA significantly reverse the deterioration of eGFR with P = 0.05. Next, to investigate whether VLP or LP is better for the CKD patients, we conducted subset analysis by comparing patients’ eGFR with KA+VLP (1.2.1) or KA+LP (1.2.2) as shown in Figure 3. Next, to investigate the optimal treatment timing amongst different CKD stages, we compared the role of kA on eGFR by assigning two subsets with eGFR < 18 (1.3.1) and eGFR > 18 (1.3.2.) as shown in Figure 4.
Q3: No statistically significant effects were observed in albumin or cholesterol levels with KA supplementation. Based on this result, the authors conclude that KA supplementation has no significant effect on patients’ nutritional status. Further discussion about the percent fat and carbohydrate content of VLP and LP diets and their effect on cholesterol level (after adjusting for statin use) and cardiovascular risk, seems warranted.
Ans3: Thanks for your comments on this important issue which we should address more. The use of cholesterol and albumin to represent the nutrition status may be arbitrary and up for debate (Ikizler T Clin J Am Soc Nephrol 9, 1375-1377, 2012; Nature Reviews Nephrology volume 7, pages 369–384 (2011)). It could be true that using a comprehensive set of parameters, such as body mass, muscle mass, dietary intake, and nutritional scoring systems, to evaluate the nutritional status instead of simply albumin/cholesterol is a better,more meaningful and more holistic approach (Am J Kidney Dis. 2009 Feb;53(2):298-309.). However, there are still arguments in favour of simply using albumin as a nutritional marker in assessing kidney diseases (Gama-Axelsson T et al Clin J Am Soc Nephrol 7, 1446-1453, 2012). In our opinion, detailed parameters, such as the percent of fat, and carbohydrate content, were not shown in most of the studies that we allocated. Therefore, it may be difficult to analyse such parameters using meta-analysis considering limited number of included studies. Further, we believe that by simply monitoring albumin and cholesterol in CKD patients is a more practical way in daily clinical practice to assess CKD patients’ nutritional status particularly in medical-lacking or developing countries. We addressed this problem in the discussion section on page 14 line 11-23
page 14 line 11-23……. Last but not least, the use of cholesterol and albumin to represent the nutrition status may be arbitrary and up for debate (Ikizler T Clin J Am Soc Nephrol 9, 1375-1377, 2012; Nature Reviews Nephrology volume 7, pages 369–384 (2011)). It could be true that using a comprehensive parameter, such as body mass, muscle mass, dietary intake, and nutritional scoring systems, to evaluate the nutritional status instead of simply albumin/cholesterol is a better and more meaningful and more holistic approach (Am J Kidney Dis. 2009 Feb;53(2):298-309.). However, there are still arguments in favour of simply using albumin as a nutritional marker in assessing kidney diseases (Gama-Axelsson T et al Clin J Am Soc Nephrol 7, 1446-1453, 2012). In our opinion, detailed parameters, such as the percent of fat, and carbohydrate content, were not shown in most of the studies that we allocated. Therefore, it may be difficult to analyse such parameters using meta-analysis considering limited number of included studies. Further, we believe that by simply monitoring albumin and cholesterol in CKD patients is a more practical way to assess CKD patients’ nutritional status particularly in medical-lacking or developing countries.
Q4: How did the short duration non-RCT study by Hecking et al (6 weeks), and the study with very low number of subjects, by Bernhard et al (N=6 per group), pass quality criteria for inclusion?
Ans4: We are appreciated with your comments. As for the first specific question, we allocated studies led by Hecking et al despite of their short period of treatment time (6 weeks) compared to other included studies simply because this study met our inclusion criteria, shown in page 2 line 21-24. Further, it was believed that the study could probably represent the short-term effects of KA on CKD patients although we did not further analyse the effects of treatment time in this study.
page 2, line 21-24……..”Studies that met all of the following criteria were included in this meta-analysis: (1) randomized controlled trials, prospective cohort and case-control studies; (2) CKD patients with reported estimated glomerular filtration rate (eGFR) data for various treatments; (3) intervention that compared KA with a low protein diet or very low protein against placebo.’’
As for the second specific question, we believe that one of the aims in conducting meta-analysis is to resolve the problems of limited statistical power due to small sample size in each individual studies. Therefore, given the small sample size in the study led by Bernhard et al, we still allocated this study to conduct meta-analysis. In our previous published papers, we allocated small-sample-size studies to conduct meta-analysis as well (Chest. 2018 May;153(5):1201-1212. AND PLoS One. 2017 Dec 14;12(12):e0188975.).
Q5: Please provide more details about the dose and type of ketoanalogues and dietary protein amounts, in g/kg/d, used in each study. Age, gender and race distribution of the subjects studied would be very useful to include in table 1.
Ans5: We revised the table 1 to incorporate the important baseline characters mentioned above.
First Author, Year | Country | Treatment/Contol | Sample size: Control (M,F) | Treatment duration | Study Design | Age (T/C) |
Bellizzi, 2006 [13] | Italy | KAVLP(1 pill/5kg/d)/FD | 30/23 | 3-6 M | Non- RCT | 58.0±16.1/56.3±15.6 |
Bernhard, 2000 [14] | France | KALP (1 pill/5kg/d) /P | 6 (4,2)/6 (6,0) | 3 M | RCT | 49.5±7.0/39.0±5.8 |
Feiten, | Brazil | KAVLP (1 pill/5kg/d) /P | 12 (7,5)/12 (8,4) | 4 M | RCT | 49.7±11.3/43.9±16.3 |
Garneata, 2016 [7] | Romania | KAVLP (1 pill/5kg/d)) /P | 104/103 | 12 M | RCT | NA |
Hecking, 1980 [16] | Germany | KALP (1.05g/10kg/d) /P | 15 (7,8)/15 (7,8) | 6 W2 | Non-RCT | 43.7±12.6/43.7±12.6 |
Klahr,1994 (HBP) [17] | USA | KAVLP (0.28g/kg/d) /P | 61/62 | 18-45 M | RCT | NA |
Klahr,1994 (LBP) [17] | USA | KAVLP ((0.28g/kg/d)) /P | 65/67 | 18-45 M | RCT | NA |
Malvy, 1999 [18] | France | KAVLP (0.17g/kg/d) /P | 25 (11,14)/25 (10,15) | 3 M | RCT | 53.6±11.0/56.0±14.0 |
Milovanova, 2018 [19] | Russian Federation | KALP (0.1g/kg/d) /P | 42/37 | 14 M | RCT | NA |
Mircescu, 2007 [20] | Romania | KAVLP (1 pill/5kg/d) /P | 26 (17,9)/19 (11,8) | 15 M | RCT | 55.0±12.7/53.6±11.0 |
Prakash, 2004 [21] | India | KAVLP (1 pill/5kg/d) /P | 18 (10,8)/16 (7,9) | 9 M | RCT | 52.8±14.1/55.9±17.6 |
Qiu, 2012 [22] | China | KAVLP (1 pill/5kg/d) /P | 12/11 | 52 M | RCT | 63.0±8.9/61.60±9.67 |
Teplan, 2008 [23] | Czech | KAVLP (0.1g/5kg/d) /P | 66/65 | 36 M | RCT | 52±7/52±7 |
Q6: For improved consistency, consider presenting KA vs control data in the same order in tables/figures throughout the manuscript. Figures 5, 6, 10, and 12 are different to the rest (controls presented before KA treated subjects in the tables), unless there is another rational behind the different format.
Ans6: Thanks for your suggestions in the format consistency. The reason that the data columns in Figures 5, 6, 10, and 12 were oppositely placed is that items in those figures showed contrary trend with the others. For example, the trend of eGFR alteration in figure 4 will just be opposite to creatinine alteration in figure 5. That is, if eGFR goes up, creatinine should go down based on the MDRD equation (Nephron 1976;16:31–41). In order to make sure that “Favour [KA]” is always on the right side of the forest plot and “Favour [control]” is always on the left side of the forest plot, we intentionally placed the data columns of creatinine (Fig. 5), BUN (Fig. 6), phosphorous (Fig. 10), and PTH (Fig. 12) in an opposite order compared to other figures.
Similar formatting can also be seen in the Figure 2 of this paper (Int Urol Nephrol. 2016 Mar;48(3):409-18).
Reviewer 3 Report
Even if diet and early time to introduce a regimen in renal diseases is of paramount importance , this study does not answer if dietary KA plus low or very low proteins are effective. Moreover to consider blood albumin as an index of malnutrition even if widely adopted is also debated (See Gama-Axelsson T et al Clin J Am Soc Nephrol 7, 1446-1453, 2012; and Ikizler T Clin J Am Soc Nephrol 9, 1375-1377, 2012). Finally, referred to the sentence at page 14 "Therefore, we believe that early KA intervention is effective in reversing CKD-MDB, but the statistical difference is only seen when the severity of CKD-MDB is high" the referee wonders if KA efficacy in highly compromised GFR patients might be clinically useful if the renal role is largely lost.
Author Response
Reviewer #3
Q1: English language and style
Ans1: Our paper has received English editing by Editage before submitting. The working number was RTSRF_26.
Q2: Even if diet and early time to introduce a regimen in renal diseases is of paramount importance, this study does not answer if dietary KA plus low or very low proteins are effective.
Ans2: We are appreciated with your comments. We mentioned the overall effects of KAVLP/KALP on eGFR on page 5, line 15-16, suggesting KA plus low or very low proteins are effective.
page 5, line 15-16……”Overall, KA significantly reversed the decrease of eGFR (Mean difference (MD) = 2.75, 95% CI = [0.71, 4.79], P = 0.008) in CKD patients (Figure 2).”
We also revised some contents in our abstract to highlight the overall effect of KA on eGFR as shown in page 1, line 20-23.
page 1, line 20-23……. A restricted protein diet supplemented with ketoanalogues (RPKA) was found to significantly delay the progression of CKD (p = 0.008), particularly in patients with an estimated glomerular filtration rate (eGFR) > 18 ml/min (p < 0.0001).
Q3: Moreover, to consider blood albumin as an index of malnutrition even if widely adopted is also debated (See Gama-Axelsson T et al Clin J Am Soc Nephrol 7, 1446-1453, 2012; and Ikizler T Clin J Am Soc Nephrol 9, 1375-1377, 2012).
Ans3: Thanks for your comments on this import issue which we should address more. The use of cholesterol and albumin to represent the holistic body nutrition status may be arbitrary and up for debate (Ikizler T Clin J Am Soc Nephrol 9, 1375-1377, 2012; Nature Reviews Nephrology volume 7, pages 369–384 (2011)). It could be true that using a comprehensive set of parameters, such as body mass, muscle mass, dietary intake, and nutritional scoring systems, to evaluate the nutritional status instead of simply albumin/cholesterol is a better, more meaningful and more holistic approach (Am J Kidney Dis. 2009 Feb;53(2):298-309.). However, there are still arguments in favour of simply using albumin as a nutritional marker in assessing kidney diseases (Gama-Axelsson T et al Clin J Am Soc Nephrol 7, 1446-1453, 2012). In our opinion, detailed parameters, such as the percent of fat, and carbohydrate content, were not shown in most of the studies that we allocated. Therefore, it may be difficult to analyse such parameters using meta-analysis considering limited number of included studies. Further, we believe that by simply monitoring albumin and cholesterol in CKD patients is a more practical way in daily clinical practice to assess CKD patients’ nutritional status particularly in the medical-lacking or developing countries. We add this issue in the discussion section on page 14 line 11-23
page 14 line 11-23……. Last but not least, the use of cholesterol and albumin to represent the nutrition status may be arbitrary and up for debate (Ikizler T Clin J Am Soc Nephrol 9, 1375-1377, 2012; Nature Reviews Nephrology volume 7, pages 369–384 (2011)). It could be true that using a comprehensive parameter, such as body mass, muscle mass, dietary intake, and nutritional scoring systems, to evaluate the nutritional status instead of simply albumin/cholesterol is a better and more meaningful and more holistic approach (Am J Kidney Dis. 2009 Feb;53(2):298-309.). However, there are still arguments in favour of simply using albumin as a nutritional marker in assessing kidney diseases (Gama-Axelsson T et al Clin J Am Soc Nephrol 7, 1446-1453, 2012). In our opinion, detailed parameters, such as the percent of fat, and carbohydrate content, were not shown in most of the studies that we allocated. Therefore, it may be difficult to analyse such parameters using meta-analysis considering limited number of included studies. Further, we believe that by simply monitoring albumin and cholesterol in CKD patients is a more practical way in daily clinical practice to assess CKD patients’ nutritional status particularly in the medical-lacking or developing countries.
Q4: Finally, referred to the sentence at page 14 "Therefore, we believe that early KA intervention is effective in reversing CKD-MDB, but the statistical difference is only seen when the severity of CKD-MDB is high" the referee wonders if KA efficacy in highly compromised GFR patients might be clinically useful if the renal role is largely lost.
Ans4: We are appreciated your invaluable suggestions in highlighting this salient issue. We believe that controlling CKD-MBD is still clinical useful despite the loss of renal function. Several studies found that high serum phosphorus and higher PTH were associated with higher cardiovascular adverse outcomes and mortality in hemodialysis patients (J Am Soc Nephrol. 2005 Jun;16(6):1788-93.; AND Kidney Int. 2005 Mar;67(3):1179-87.; AND Am J Kidney Dis. 2008 Sep;52(3):519-30). Our past paper investigating CKD-MBD biomarkers on mortality in peritoneal dialysis patients also found similar association (Scientific Reports volume 7, Article number: 33 (2017)). Therefore, according to the above studies it seems to be clear that having good control of CKD-MBD can prolong the survival of advanced CKD patients.

Round 2
Reviewer 3 Report
Many efforts have been made to justify criticisms and the revised version could be acceptable even if caution must be due to studies that are lacking of detailed clinical data. In my opinion, comparative analysis is not appropriate when heterogeneous data and different methods are reported.
Author Response
Q1: Many efforts have been made to justify criticisms and the revised version could be acceptable even if caution must be due to studies that are lacking of detailed clinical data. In my opinion, comparative analysis is not appropriate when heterogeneous data and different methods are reported.
A1: We are appreciated your great comments on this critical issue. We mentioned this concern in the Discussion on page 15, line 19-20. However, it is not denying that some researchers argued that the advantages of including observational studies into a meta-analysis could outweigh the disadvantages (Am J Epidemiol. 2007 Nov 15;166(10):1203-9. Epub 2007 Aug 21.). In order to further highlight this important issue, we slightly revised our manuscript.
page 15, line 19-22……Second, we included 12 studies comprising 10 RCTs and two non-RCTs,. which may lead to increased heterogeneity and uncontrolled bias although some arguments state that the advantages of including observational studies into a meta-analysis could outweigh the disadvantages [56].